# A Review of Polylactic Acid as a Replacement Material for Single-Use Laboratory Components

**DOI:** 10.3390/ma15092989

**Published:** 2022-04-20

**Authors:** Brian Freeland, Eanna McCarthy, Rengesh Balakrishnan, Samantha Fahy, Adam Boland, Keith D. Rochfort, Michal Dabros, Roger Marti, Susan M. Kelleher, Jennifer Gaughran

**Affiliations:** 1School of Biotechnology, Dublin City University, D9 Dublin, Ireland; rengesh.balakrishnan@dcu.ie; 2I-Form, Advanced Manufacturing Research Centre, Dublin City University, D9 Dublin, Ireland; eanna.mccarthy@dcu.ie; 3Office of the Chief Operations Officer, Dublin City University, D9 Dublin, Ireland; samantha.fahy@dcu.ie; 4Grain-4-Lab, Dublin City University, D9 Dublin, Ireland; adam.boland@dcu.ie; 5School of Nursing, Psychotherapy and Community Health, Dublin City University, D9 Dublin, Ireland; keith.rochfort@dcu.ie; 6School of Engineering and Architecture of Fribourg, HES-SO University of Applied Sciences and Arts Western Switzerland, CH-1700 Fribourg, Switzerland; michal.dabros@hefr.ch (M.D.); roger.marti@hefr.ch (R.M.); 7School of Chemical Sciences, Dublin City University, D9 Dublin, Ireland; susan.kelleher@dcu.ie; 8School of Physical Sciences, Dublin City University, D9 Dublin, Ireland; jennifer.gaughran@dcu.ie

**Keywords:** bioplastics, polylactic acid, 3D printing, biodegradable polymers, lab consumables

## Abstract

Every year, the EU emits 13.4 Mt of CO_2_ solely from plastic production, with 99% of all plastics being produced from fossil fuel sources, while those that are produced from renewable sources use food products as feedstocks. In 2019, 29 Mt of plastic waste was collected in Europe. It is estimated that 32% was recycled, 43% was incinerated and 25% was sent to landfill. It has been estimated that life-sciences (biology, medicine, etc.) alone create plastic waste of approximately 5.5 Mt/yr, the majority being disposed of by incineration. The vast majority of this plastic waste is made from fossil fuel sources, though there is a growing interest in the possible use of bioplastics as a viable alternative for single-use lab consumables, such as petri dishes, pipette tips, etc. However, to-date only limited bioplastic replacement examples exist. In this review, common polymers used for labware are discussed, along with examining the possibility of replacing these materials with bioplastics, specifically polylactic acid (PLA). The material properties of PLA are described, along with possible functional improvements dure to additives. Finally, the standards and benchmarks needed for assessing bioplastics produced for labware components are reviewed.

## 1. Introduction

Polymers are materials made of long chains of organic molecules. Their low cost, durability, and ease of manufacture has made them attractive materials for many applications, with synthetic polymers (plastics) becoming ubiquitous materials in packaging, clothing, and other products. The most common plastics are petroleum-based and non-biodegradable. By 2015, 6300 Mt of plastic waste had been generated globally, with just 9% being recycled [1]. Single-use plastic laboratory consumables alone have been estimated to generate 5.5 Mt in 2014 [2]. In bioprocessing, single-use disposable equipment has reached ≥85% of pre-commercial production by 2018, and is increasingly being incorporated in commercial manufacturing [3]. Single-use plastic equipment can arrive pre-sterilised and can be disposed of once contaminated, simplifying the facilities and processes needed for research labs or production facilities [4]. This allows for faster establishment or augmentation of production lines at low capital cost, and as such can be more cost effective than reusable glass or stainless-steel equipment at all but the largest scales [3,4,5]. However, it increases the dependency on fossil fuel sources and the volumes of non-degradable waste generated.

Based on current trends, Geyer et al. [1] report an estimated 12,000 Mt of plastic in waste landfills by 2050. This large accumulation of non-degradable waste along with penetration into ocean, freshwater, and terrestrial habitats, present a serious environmental concern [6]. These concerns drive interest in the development of eco-friendly degradable polymers, such as bioplastics.

The term bioplastics is used to describe two (overlapping) classes of polymer: those that are biodegradable, and those that are derived from biomass [7]. The production of biodegradable or recyclable polymers using biomass waste materials from industries like agriculture and food can provide multiple benefits to sustainability; making use of material that would otherwise go to waste, reducing dependence on fossil fuels, and reducing or eliminating the environmental impacts of disposal of the plastics.

## 2. Common Labware Polymers and Production

Plastic labware is made of a range of polymers depending on the properties that are needed for a given item (flexibility, chemical resistances etc.). A summary of common polymers [8] used for typical items and their general properties is presented in Table 1.

The most common polymers used for single-use labware will be discussed in more detail in the review including: PS, PETG, PC and PP.

### 2.1. Polymer Properties

#### 2.1.1. Polystyrene

For single-use items like petri dishes and pipette tips, polystyrene is a commonly used material due to its low cost, transparency, rigidity, chemical inertness, chemical stability, and ability to be functionalised [9]. Polystyrene is a thermoplastic polymer, made from the aromatic hydrocarbon monomer styrene which is derived from fossil-fuels [10,11]. Polystyrene is sufficiently chemically resistant for single-use with acids, bases, alcohols, oxidising agents, aqueous solutions, and detergents/surfactants [12]. The material is not suitable for autoclave sterilisation, but is often sold pre-sterilised using ethylene oxide gas sterilization or gamma irradiation [8]. The properties for standard polystyrene are shown in Table 2. Polystyrene is not biodegradable, though it can be recycled [13,14].

#### 2.1.2. Polyethylene Terephthalate G Copolyester

Polyethylene Terephthalate G Copolyester (PETG) is used for some single-use labware items such as bioprocessing bottle assemblies [18], and disposable Erlenmeyer flasks [19]. PETG is a copolymer of the commonly used plastic Polyethylene Terephthalate (PET). PET is a thermoplastic polymer made from the hydrocarbon monomer ethylene terephthalate which is derived from fossil fuels [20]. PET is not biodegradable but it can be recycled [20]. The general properties for PET and PETG are shown in Table 3. PETG has a lower melting temperature and better flexibility than PET, and does not crystallise, which can lead to opacity and brittleness in PET when exposed to heat. PETG’s lower melting point makes it unsuitable for autoclaving, which is a limitation for reusable labware, but can be pre-sterilised with radiation or compatible chemicals for single-use items [8]. This has made PETG popular for single use vessels and as an input material for polymer filament based additive manufacturing.

#### 2.1.3. Polycarbonate

Polycarbonate is also used in single-use bottle assemblies [18,24]. It may also be used in disposable pipette tips [25], and Erlenmeyer flasks [19]. Polycarbonates are a class of aliphatic and aromatic polymers [26,27]. They are a common engineering material due to their attractive properties such as high impact resistance, heat resistance, flame retardancy, and good transparency [28]. Typical polycarbonates are oil based and degrade poorly; however, there have been advances in applying bacteria or enzymes to degrade the material, and developing sustainable-sourced and biodegradable polycarbonates [28,29]. Polycarbonate is transparent, strong, rigid, and autoclavable [8]. The general properties are shown in Table 4. Polycarbonate is sufficiently chemically resistant for single-use with dilute or weak acids, dilute or weak bases, oxidising agents, aqueous solutions, and detergents/surfactants [12].

#### 2.1.4. Polypropylene

Polypropylene is used for some disposable lab consumable applications such as centrifuge tubes [33], disposal bags [34], and pestles [35]. Polypropylene is a petroleum-derived thermoplastic polyolefin, produced by polymerising propylene [36]. The general properties are shown in Table 5. Polypropylene is sufficiently chemically resistant for single-use with acids, bases, oxidising agents, aqueous solutions, and detergents/surfactants [12].

### 2.2. Plastic Labware Production Techniques

Plastic manufacturing processes have been developed to cover a wide range of requirements for material types and end-user applications. The choice of techniques is usually driven by several of factors including: design considerations, product development stage, material choice, cost and instrumentation availability. The most common techniques employed are 3D printing, injection moulding and blow moulding, which can also be applied to PLA-based components.

Blow and injection moulding are a popular method of plastic processing and known to produce high volume and mass production of commercial products including labwares. Blow moulding favors hollow, single-piece parts that can be flexible, structural, or can hold a fluid, such as centrifuge vials etc [5]. Injection molding has several advantages that include repeatable high tolerances, low labor cost, minimal loss and minute treatments for finished goods [39]. On the downside, they require monitoring from vendors, manufacturers and suppliers and have high costs initially in the manufacture of the part’s mould (a.k.a. die). Therefore, the costs for final goods and prototypes produced will be expensive for unique or unusual objects and designs, and thus injection moulding is only really cost effective for mass production of parts. Also, researchers may need to buy raw materials in bulk without being able to obtain one piece for a quick evaluation [40]. Sang et al. [41] looked at the oscillation shear flow during the packing stage of an injection moulding cycle. They made special injection-molded PLA parts with different thicknesses and crystallinity of skin layers by controlling shear durations and oscillation frequencies. For the part of the 2000 μm thick layer with 50% crystallinity, the heat distortion temperature and Vicat softening temperature reach 96.6 and 159.3 °C, respectively. Furthermore, the Young’s modulus rises a lot with the thickness and crystallinity of the material.

3D-printing is a method frequently used for the manufacture of parts from plastics or metals, by the deposition of layer upon layer of material to build a three dimensional shape [42,43,44,45]. A benefit of 3D printing is the economy of scale. Whereas techniques like injection moulding are less costly for mass production of parts, 3D printing has become competitive for prototyping, bespoke parts or smaller production runs, offering customisations as per consumer needs [46]. Though not used widely as a technique for labware manufacture, the open-source nature of the technique has allowed for the scientific and engineering labs to build items required within the lab, including micropipettes, microscopes, magnetic stirrers and syringe pumps. Replacement parts for the International Space station can now be manufactured in situ from plastics or metals [47]. A significant benefit of 3D printing is its compatibility with a wide range of materials, making it an excellent technique for testing the capabilities of new materials. It should be noted that there are some significant limitations in the use of 3D printing for labware production including limited resolution, poor optical clarity, solvent and chemical compatibility issues and slow throughput. In particular, the slow nature of the printing speed makes it a poor choice for mass production, though significant advancements in the technology have led to improvements in many of these areas [48].

## 3. Polylactic Acid (PLA)

Sustainability needs have driven interest in polymers which are degradable, recyclable, and/or derived from eco-friendly input materials. Biomass-based polymers are one option with many benefits to sustainability: reducing dependence on fossil fuel extraction; produced from renewable resources; may make use of waste materials; and better biodegradability [49]. Polylactic Acid (PLA) is one of the most commonly used bioplastics, in 2021 it was reported to hold the largest market share worldwide [50] for biodegradable bioplastics manufacturing capacity. PLA’s manufacturing capacity accounted for 42% of the total biodegradable, bioplastics production worldwide, with its nearest rival polyhydroxyalkanoates (PHA) [7,51] accounting for only 4% of worldwide production [50].

PLA can be produced from starch using fermentation by microorganism to create the monomer lactic acid, which is polymerised to form PLA [49]. As such, it can be produced from renewable sources. The reduction of CO_2_ emissions is one of the most advantageous aspects of PLA manufacturing when compared to alternative hydrocarbon-based polymers. Carbon dioxide is thought to be the most significant contributor to global warming and climate change. PLA has the potential to release less greenhouse emissions than rival hydrocarbon-based polymers because CO_2_ is absorbed from the air when maize is cultivated [52]. On the other hand, agricultural waste material which may otherwise go unutilised can be used to produce the lactic acid, such as dairy waste, cottonseed, tobacco waste, wheat straw, corn cobs, coffee pulp, food waste, stillage, and used brewer’s grain [53]. The general properties of PLA are shown in Table 6.

### 3.1. Mechanical Properties of PLA

Typical PLA is a brittle material [49]. Anderson et al. [56] compare PLA’s properties to those of two polymers commonly used in single-use labware, polystyrene (PS) and PET (see Table 7). Brittle PLA’s impact strength is similar to polystyrene, with tensile strength and modulus that are more comparable to PET. However, the mechanical properties of PLA can range from soft, elastic materials to stiff rigid ones, depending on several physical factors such as the molecular weight and crystallinity, and the use of polymer blending or composite additives.

Perego et al. [58] report on the effect of molecular weight and crystallinity on the mechanical properties of several PLA variants; Poly(L-lactide) (PLLA) and Poly(D,L-lactide) (PDLLA). L-lactide and D-lactide are two stereoisomers of lactic acid (see Figure 1), which can lead to differing properties in the PLA produced from them [49]. Annealed PLLA samples were also characterised to investigate the effect of crystallinity. The properties are shown in Table 8. Molecular weight was shown to have a much stronger effect when the crystallinity was higher. Impact resistance was also strongly influenced by crystallinity.

### 3.2. Thermal Properties of PLA

The isomer composition can also affect the thermal properties of PLA. Ahmed and Varshney, investigated samples of PLA derived from L-lactide, D-lactide, or both with varied molecular weights [59]. Avinc and Khoddami, [60] illustrate the molecular configurations of PLA with differing isomer compositions (see Figure 2). The results for the varied PLA samples are shown in Table 9. The authors note that the melt and glass temperatures (*T_m_* and *T_g_*, respectively) tend to increase with the number average molecular weight (*M_n_*) irrespective of isomer. A range of glass temperatures (*T_g_*) can be achieved, with low values giving easier processability and higher values allowing higher operating temperatures for parts produced.

### 3.3. Mitigating PLA’s Functional Limitations

Standard PLA has some limitations which may make it unsuitable for certain applications. The low glass and melt temperature make it unsuitable for high temperature operations [59]. Sin et al. [49] report that PLA is suitable for room temperatures but seldom used for higher temperature applications as it begins to lose structural integrity at ≥60 °C, making it unsuitable for PCR tests, boiling water, and thermal sterilisation. PLA’s ductile-brittle temperature, the temperature below which polymers are fully brittle, is −47 °C [61].

Janorkar et al. [62] report reductions in the molecular weight of PLA after exposure to a UV sterilisation lamp, which may indicate standard PLA is unsuitable for UV sterilisation. The UV resistance of PLA can be improved by some additives [63,64,65]. Ho et al. [65] found the addition of bamboo charcoal particles to PLA mitigated deterioration of the mechanical properties after UV exposure. Man, et al. [63] report improved UV resistance with the addition of rutile TiO_2_ in spin coating and extrusion-injection processing. Zhou et al. [66] found that PLA-TiO_2_ nanocomposite membranes could be safely treated with UV-assisted cleaning. Cao et al. [67] coated TiO_2_ nanoparticles with SiO_2_ then D-lactide based PLA to create double shell particles, and used this to reinforce L-lactide based PLA. The reinforced PLA was able to maintain its mechanical properties by >90% after 72 h of UV exposure. The initial mechanical properties were also enhanced by the reinforcement, with the tensile strength being 49% higher.

### 3.4. Biological Compatibility Requirements of PLA

When considering a material to interface with a biological system, the biocompatibility of the material is critical in preserving the systems biological integrity. Biocompatibility is a measure of a material’s ability to come into contact with a biological system without eliciting detrimental responses those which elicit little to no effect can be thus considered as inert and biocompatible in nature [68]. This is of particular importance from a standardisation perspective as the choice of alternatives to petroleum-based, non-biodegradable plastics increases steadily [50]. In producing a material, elements such as process contaminants, residues, leachables, and products of degradation may comprise a percentage of the material and in-turn may potentially influence certain biological interactions differently to that intended with the original material. In that way, any and all compounds and additives which may be present and/or intentionally added to improve the properties of these alternatives, must be assessed in terms of their biocompatibility for not just the sample in which it will come into contact with, but also the user [51]. In that way, biocompatibility testing is incredibly important in order to determine whether the alternative material is question is indeed fit for purpose, and functions in exactly the same way as those materials which it is replacing. As such, all potential materials must meet criteria developed by the International Organisation for Standardisation (ISO) in which lab materials made from said alternatives must meet a series of standards which are recognised by regulatory authorities all over the world [69].

Depending on the purpose of the material, the testing methods use to evaluate the biocompatibility can vary widely. These tests, with methodologies spanning in vitro and in vivo, can vary in turnaround time from days to several months depending on the requirement for the specific test data, though some tests may not be required depending on the application [69]. Biological properties such as genotoxicity, hemocompatibility, sensitisation, irritation, implantation and system toxicity are among the indices typically examined with respect to lab materials, though the most common assays used are those which identify the cytotoxicity of the material. Direct contact cell culture assays which evaluate the impact of the material on cell adhesion, cell activation, and/or cell death are used extensively in biocompatibility studies of novel materials [70], as well as extractions in which the leachable materials from the test material are harvested in response to different solvents, and analysed for potentially harmful chemicals or cytotoxic molecules [71]. The aforementioned are mandatory for all lab-based product evaluation programs run by national regulatory bodies, with additional tests such as material properties (chemical, mechanical and thermal) with respect to the potential application also required. Moreover, considerations must also made for instances of misuse of the material/product from both biological and material perspectives [69].

Should a material meet the designated standards, and indeed be deemed biocompatible and fit for its intended purpose, the impact of sterilisation on the material should also be considered. Methods of sterilisation can vary in nature, and as all lab materials must be sterilised before coming into contact with a biological system/host, the degree of stress imparted by a means of sterilisation should be taken into consideration. Single-use plastics only need to tolerate one cycle of sterilization; however, multiple-use products can be subjected to several cycles in their life-time and are required to be able to withstand such without any change in their functional and mechanical properties [72]. As this is deemed an integral process which all lab materials are subjected to prior to use, the impact of sterilisation techniques on prospective replacement materials such as PLA are determined very early in the characterisation process.

Sterilisation is important for many plastics applications like medical devices, food packaging, and lab ware. Farah et al. [73] summarise non-UV sterilisation and their advantages and disadvantages, shown in Table 10.

### 3.5. Solvent Interaction with PLA

Generally, PLA is not dissolved in water, selective alcohols, and alkanes; however, amorphous PLA is highly soluble with organic solvents [49]. Hansen, [74] reports the solubility parameters for several solvents at 25 °C, shown in Table 11. *δ_d_* is the dispersion solubility parameter, *δ_p_* is the polar solubility parameter, *δ_h_* is the hydrogen solubility parameter, and *δ_t_* is the total solubility parameter. Close values in the total solubility parameters for two materials (Sin et al. [49] specify <2.5 difference in *δ_t_*) indicate solubility. Agrawal et al. [75] calculated solubility parameters for standard PLA with a number of methods, shown in Table 12. It is indicated that standard PLA is expected to dissolve in acetone, benzene, chloroform, 1-4 dioxane, 1-3 dioxolane, ethyl acetate, furan, isoamyl alcohol, methylene dichloride, methyl ethyl ketone, tetrahydrofuran, toluene, and xylene. Nampoothiri et al. [76] note that PLA is only weakly soluble in acetone, ethyl benzene, toluene, and tetrahydrofuran at room temperature but can be readily dissolved with heating. However, high crystallinity PLLA can resist acetone, ethyl acetate, and tetrahydrofuran [49,76].

### 3.6. Effect of Temperature on Leachables from PLA

Mutsuga et al. [79] reported on the leaching of lactic acid, lactide, and oligomers from PLA at different temperatures. PLA sheets procured from different manufacturers were placed into glass tubes with 100 mL water, and water with 4% acetic acid or 20% ethanol, and the presence of migrated compounds was measured with liquid chromatography/mass spectrometry after several fixed periods. They note that migration into the mixture was present but low for 20 or 40 °C, but that at 60 °C or higher there are significant migrant levels due to the decomposition of the PLA. Lactide migration levels were raised to 0.24 mg.cm^2^ at 60 °C, 0.64 mg.cm^2^ at 80 °C, and 4.12 mg.cm^2^ at 95 °C and only at that higher temperature oligomers were identified at 1.98 mg.cm^2^. After the migration test at 95 °C, the sample turned cloudy. It was observed that LA was calculated to be formed at 95 °C from lactide produced by PLA degradation. The authors note that migrant levels were worse for samples with a higher ratio of D-lactide. For the lower temperatures, the amount of lactic acid is lower than those present in some common food ingredients, making PLA suitable for food packaging applications [49,79]. However, the leaching of material may interfere with sensitive lab applications, particularly those involving higher temperature chemical processes.

### 3.7. Additives to PLA

Plasticisers may be used to increase the ductility of brittle polymers. For PLA, the glassy brittle polymer can be plasticised using its own monomer (lactic acid/lactide) to increase its flexibility. Sinclair reports the tensile properties for different percentages of lactide plasticiser [80]. The data for different percentage of plasticiser illustrate the broad range of properties that can be obtained, and some similarities to conventional thermoplastics can be noted. This allows PLA to compete as a sustainable packaging material; however, the authors note that degradation is increased with increasing plasticiser [80].

In addition to using plasticisers to improve ductility, other additives may be used with PLA to improve or tailor other properties. Polymer blends are another approach for achieving desired properties. PLA can be blended with several other polymers. For example, PLA can be blended with rubbery polycaprolactone (PCL) to increase elongation at break with reduced tensile strength and stiffness [81,82]. Impact modifiers can be added to improve the impact strength, and nanoscale clay particles can be added to improve stiffness [83]. Additives may be mixed to achieved the desired balance of properties, as illustrated in Figure 3 [83].

Additives can also be used to improve the heat resistance. Heat resistance is evaluated by several methods characterizing how the material’s resistance to deformation changes with temperature. The Heat Deflection Temperature (HDT) for PLA, defined as the temperature at which a defined specimen deflects 250 µm under a specified load and heating rate, is 55 °C, and the Vicat Softening Temperature (VST) for PLA, defined as the temperature at which a specimen is penetrated to 1 mm by a flat ended 1 mm^2^ area pin under a specified load and heating rate, is 65 °C [84]. Heat resistance can be improved by increasing the crystallinity (using nucleating agents or processing strategy), by polymer blending, or by other reinforcement additives. Using nucleation additive dibenzoylhydrazide with PLLA, Kawamoto et al. [85] were able to achieve a HDT of 124 °C. The stiffness and Izod impact strength were also improved, 4.1 GPa and 7.9 kJ m^−2^, respectively. In terms of processing, Tábi et al. [86] produced PLA samples using injection molding with different mold materials. Additively manufactured epoxy-based molds were able to deliver higher crystallinity due to their slowing cooling rates compared to steel molds. Using PLA containing nucleation agents (talc and polyethylene glycol (PEG)) with both mold types at room temperature, the VST was improved from 60–65 °C for the steel molds, to 118–124 °C for the epoxy-based molds. The strength of the interfacial bond between adjacent layers dictates the mechanical characteristics of 3D components manufactured using the fused filament fabrication (FFF) process. When PEGs are added to FFF-printed PLA components, the interlayer bond strength is increased, lowering the mechanical anisotropy from 32% for pristine PLA parts to 16% for PLA/PEG parts. Additionally, PEGs with a molecular weight greater than 8000 g/mol have a significant impact on the mechanical characteristics of PLA components [87]. Hriţuc et al. [88] looked into the 3D printing processes that use polylactide (PLA) wire to make parts in a wide range of shapes and sizes. The fused deposition modelling process had some differences between the desired dimensions and the real dimensions that were made. Taguchi models used in the study show that in the case of tubular parts made of PLA, printing speed and plate temperature have the biggest impact on the height and diameter of 3D printing.

PLA can also be used as a matrix in composite materials. Composite materials are made up of two or more different component materials with advantageous properties. For polymers, reinforcement with stiff fibres is a popular approach. Fibre-Reinforced Plastics (FRPs), using fibres such as glass or carbon in the polymer matrix, can provide high specific stiffness, specific strength, impact strength, and damping [89]. Their ability to deliver high strength-to-weight ratios has made FRPs popular in automotive, aerospace, and wind turbine applications [89,90,91,92,93,94,95]. Glass fibres have been used with PLA, providing increases in tensile strength, flexural strength, impact strength, and heat deflection temperature of 183%, 134%, 331%, 313%, respectively, using 30% glass fibre [96]. Carbon fibres have also been used with PLA, giving increases in tensile strength, tensile modulus, flexural strength, and flexural modulus of 73%, 438%, 53%, and 400%, respectively, with 30% carbon fibre [97]. It was reported by Agüero et al. [98] that they made and characterised green composites that used PLA-based fillers and additives that came from the linen processing industry. They showed that the waste from flax (Linum usitatissimum L.) or byproducts can be used to get renewable raw materials that can be used to make green composites for market applications like rigid food packaging and food-contact disposable items in the circular economy and bioeconomy.

#### Sustainable, Compostable PLA Additives

Sustainable composites can be achieved by combining PLA with eco-friendly additives [99,100]. Cali et al. [101] report on the use of PLA matrices with agricultural waste fillers to create composite filaments which can be used in additive manufacturing (AM). Combining an eco-friendly, biodegradable polymer with biodegradable waste materials that might otherwise go unused to improve the mechanical properties of produced parts is a boon for the sustainability of composite parts, and of additive manufacturing as a technique. The authors used the filaments to successfully produce two biomedical prototypes [101]. Similarly, Matsuzaki et al. [102] describe a method where jute fibre is fed into a heated nozzle with pure PLA filaments during the AM process, achieving modulus and strength increases of 157% and 134%, respectively, compared to pure PLA. By combining PLA with poly (*ε*-caprolactone) (PCL), a biodegradable, water insoluble polyester, López-Rodríguez et al. [103] found that an increase of PCL led to a decrease in the Young’s modulus and the tensile strength of the composite (from 56.8 MPa at 0% PCL to 12.5 MPa at 80% PCL). At a composition ratio of 80% PLA and 20% PCL this blend was found to have similar mechanical properties to PS. Composite blends of PLA and silk fibroin (SF) containing 2–10 wt.% of PLA dispersed in a SF matrix displayed an increase in Young’s modulus (from 2876 MPa at 0% SF to 3480 MPa at 90% SF), tensile strength (from 23.2 MPa at 0% SF to 28.5 MPa at 90% SF), and hydrophobicity in comparison to neat PLA [104]. The mechanical properties of blends containing poly(butylene succinate) (PBS) and PLA were investigated by Chen et al. [105] and Wang et al. [106]. PBS is typically added into a blend with PLA in order to increase the toughness of the material without compromising the biodegradability of the plastic material [107,108,109]. More recently still, Rojas-Martínez et al. [110] have published results showing that PLA blended with keratin and chitosan can be 3D printed into scaffolds. This year, Brounstein et al. [111] reported the blending of PLA with TiO_2_, ZnO, and ceramics (up to 30 wt.%) to produce antimicrobial composites. Banerjee et al. [112] evaluated the many kinds of nanoparticles employed in the manufacture of PLA nanocomposites, including nanoclay, nanocelluloses, carbon nanotubes, and graphene, covering all key processing, characterisation, and application elements. Dedukh et al. discussed new developments in the usage of composite materials manufactured from PLA in bone surgery and the use of 3D printing to create implants [113,114,115]. The usage of PLLA and PLA blends will rise as we learn more about how to adjust the mechanical response of this essential class of materials. These bioresorbable polymers have the capacity to breakdown under biomedically relevant situations. This degradation is regulated by the moleculer weight and orientations, crystallinity, and the chemical and load environment [116]. The impact of acrylated epoxidized soybean oil (AESO) addition on the mechanical, thermal, and thermomechanical characteristics of PLA components formed by injection moulding was studied by Quiles-Carrillo et al. [117] PLA components with 2.5–7.5 wt.% AESO exhibited a significant increase in elongation at break and impact-absorbed energy, but their tensile and flexural strength, as well as thermomechanical characteristics, were preserved or slightly enhanced. It also had a stronger thermal stability and a reduced crystallinity. Thus, using AESO to generate toughened PLA materials of great interest in rigid packaging, automotive, or building and construction applications might be deemed an ecologically acceptable alternative. PLA is gaining a reputation for being a flexible material, from which composites and blend materials can be produced, in order to fine tune the required physical and chemical properties. The key factor for consideration for these developments is not only the improved properties of the PLA, but also the degradation pathways of the materials and end destination of the additives.

### 3.8. PLA Industrial Synthesis Processes

The industrial production of PLA today is mainly based on the ring opening polymerization (ROP) of lactide because polycondensation of lactic acid requires rather harsh conditions, i.e., high temperatures (180–200 °C), vacuum (at least 5 mbar) and long reaction times to obtain PLA of high molecular weights [118,119]. In contrast, ROP works at rather quite mild conditions (max 130 °C, reaction times of several hours) and yields PLA with narrow PDI and high molecular weights of up to 100 kDa, which are important for reasonable mechanical properties. Metal alkoxides as for example tin(II) octoate are preferred as industrial catalysts as they give high molecular weight and no loss of optical purity [118,119]. The chiral integrity is crucial for the properties of PLA (as discussed in Section 3.2). Gupta et al. [120] reviewed the uses of PLA and its potential value in a variety of emerging technologies, including orthopaedics, drug administration, sutures, and scaffolds, and have piqued researchers’ curiosity in this innovative field. Additionally, they addressed developing PLA using a range of catalysts to meet a variety of performance needs.

Cargill Inc. was the first to industrialize the ROP process from L-lactide in the early 1990s. The required lactide is produced starting from lactic acid in a continuous process; the LA is first condensed to produce a low molecular weight prepolymer PLA from which by controlled depolymerization produces the lactide. The typical operating conditions for the reactor were residence time about 1 h, vacuum pressure 4 mbar, temperature 210 °C, and catalyst amount 0.05 wt.% tin(II) octoate in the feed. The crude lactide is separated and purified by distillation as the specifications for lactide are stringent, especially in terms of free acidity, water content, and stereochemical purity [119]. The industrial PLA synthesis is often referred to as a two-step process because lactide synthesis and ROP are combined [119,121]. As an example, as a typical ROP process, Figure 4 shows that of NatureWorks based on the original Cargill-Dow patented process. NatureWorks produces the USA with a plant with a total capacity of 150,000 t/yr. The second largest plant is the 75,000 t PLA plant in Thailand under the joint venture of Total and Corbion companies [121]. For this later process Sulzer developed together Corbion a continuous process based on the use of static mixers, called SMR™ (Sulzer Mixing Reactor, Sulzer, Winterthur, Switzerland) [122]. This reactor is characterized by a precise control of heat transfer and mixing effects, which allow a high turnover and a consistently high polymer quality. Subsequent devolatization via a static degassing technology allows the elimination of volatiles in the PLA and thus the recycling of unreacted lactide [122].

### 3.9. PLA Current Applications

PLA is used in a wide range of applications ranging domestic, engineering, agricultural, and biomedical sectors [49,123]. PLA fibres can be used in packaging, clothing, furnishings, bedding (pillows, mattresses etc.), and other applications such as hygiene products [60,124]. Lunt and Shafer, [124] noted the advantages of PLA fibres for garments: better wicking and moisture regain; more comfortable; good resilience; unaffected by UV; low flammability; lower stiffness leads to better drape/hang; good crease resistance; dyeability; and sustainability. Superior self-extinguishing behaviour and lower smoke generation compared to other polymer fibres like PET, along with greater resilience and better sustainability, make PLA attractive for furnishings such as drapes and upholstery [124]. The renewable and biodegradable nature of PLA makes it well suited for single-use applications, such as packaging and containers [59]. For example, Swiftpak produce PLA insulation bags [125] and PLA bottles EU produce PLA bottles for milk and water [126]. Common packaging polymers like PET, PVC, polyethylene, polystyrene, and polyamide are petrochemical based with poor degradability. PLA’s properties are adaptable as described above which give scope to tailor the material to a range of packaging products. The Food and Drug Administration (FDA) in America has approved PLA for use in all food packaging applications [73]. PLA has been a popular material for additive manufacturing or 3D printing [100,127,128]. The relatively low glass and melt temperature (see Table 9) make PLA easy to process with thermal methods like Fused Deposition Modelling (FDM). For rapid modelling or prototyping, many parts may be produced and discarded, so good degradability and good recyclability as a thermoplastic are other advantages to PLA. As it is non-toxic to the human body, and a bioabsorbable polymer, PLA is attractive for medical applications [73]. It may be used for sutures, dental implants, and drug-delivery devices [129].

#### PLA for Labware Applications

PLA’s biodegradable attributes makes it attractive for disposable single-use labware items; however, uptake of PLA in labware has been limited. Properties like temperature, UV, and chemical resistance may be limiting factors for this application. SP Scienceware subsidiary Bel-Art produces a range of “Earth-Friendly” spoons and sampling sticks [130,131]. Baden et al. [47] describe the use of additive manufacturing to produce labware in-house using PLA among other materials; however, the focus here was on longer term jigs and fixtures, rather than lab consumables. Gordeev et al. [132] investigated the use of AM to produce chemistry equipment from several engineering polymers including PLA. 3D printed test tubes of PP, PLA, Acrylonitrile Butadiene Styrene (ABS), and PETG were produced and characterised, and the suitability of the materials was assessed in the study. Polypropylene is the most suitable material for chemical experiments due to its high resistance to chemical reagents; PLA labware, on the other hand, has superior properties: it has almost no pores and is very tight, the material does not shrink significantly, and the material is convenient for additional mechanical post-processing. While PETG products are partially transparent, which is an undeniable benefit, the layered structure created during printing precludes the use of PETG printed labware owing to its high porosity. ABS’s limited chemical resistance severely limits its use in chemistry. As a consequence, PP and PLA are much more appropriate for printing labware than ABS or PETG. The following is a general order of the functioning of plastic materials for chemical applications: PP > PLA > ABS > PETG. The mild solvents were Et_2_O, EtOH, hexane, and H_2_O, and the aggressive solvents were acetone, MeCN, CH_2_Cl_2_, THF, toluene, and DMSO.

## 4. Labware Needs Assessment

### Standards

There are a number of ISO standards relating to common plastic laboratory wares [133]. The details for these are shown in Table 13. To-date no literature has directly indicated if PLA can meet the ISO requirements de-scribed below. However, the standards remain the framework that PLA-based components must meet to be adopted as plastic labware alternatives. The material properties of PLA described in Table 8, along with functionality improvements using additives illustrated in Table 9 suggest that targeted PLA-based labware components could be manufactured to meet several ISO standards.

ISO 6706:1981 for plastic graduated cylinders indicates that the material for plastic labware must be rigid, non-brittle, translucent or transparent, possess suitable chemical and thermal properties, and be as free as possible from defects or stresses [134]. It gives a standard for the transparency, which is that the cylinder should allow the meniscus of transparent liquids to be visible through the cylinder wall. Two tests are indicated in this standard; one to calibrate the volumetric measurements for thermal expansion of the vessel, and another examining the ionic material extracted by water at 20 °C to determine if it is within permitted limits set by the standard, based on changes to the conductivity of deionised water.

ISO 7056:1981, 7057:1981 indicates that the material for plastic labware must be rigid, non-brittle, possess suitable chemical and thermal properties, and as free as possible from defects or stresses [135,136]. ISO 7056:1981 for plastic beakers gives limits for acceptable extraction of ionic material by water at 60 °C, again based on changes to the conductivity [135]. A flexibility test is also described, a beaker is filled with 60 °C water and the pin are applied with 30 N of force for 1 min, and the percentage change in outer diameter is determined. The standard prescribes that the wall thickness and design of the beaker should be such that the change in outer diameter is <10%.

ISO 7057:1981 for plastic filter funnels also gives a testing procedure for extraction of ionic material by water at 60 °C, where a borosilicate stopper is used to allow the funnel to be filled with deionised water and held at 60 °C for 3 h, and gives a maximum acceptable change in conductivity of 200 µS/m [136]. A flexibility test is also described; again, the funnel is stoppered and filled with 60 °C water. A 1 kg weight is hung from the rim of the funnel for 1 min, and the outer diameter measured. It is specified that the percentage change in outer diameter should be no more than 5%.

ISO 12771:1997 for disposable serological pipettes specifies the material must be translucent and possess suitable chemical and thermal properties [137]. ISO 24998:2008 for single-use petri dishes specifies that the material must be microbiologically inert and transparent, and that the finished dish must be free from colour variation or discolouration, and from physical defects such as striations which could impair its use in microbiology [138]. The standard also describes a flexibility test, where a test finger applies a steady load of 4.9 N and the change in the inner diameter after 10 sec is measured. It is specified that the diameter of the lid and the diameter of the dish should not decrease by more than 1 and 2 mm, respectively. The standard also describes a test for resistance to thermal distortion. 20 ± 0.5 mL of an aqueous solution with 1.5 ± 0.2% agar at 60 ± 2 °C is poured into a petri dish, the lid placed on the dish, and the solution allowed to cool and set. The flatness is then measured in terms of the maximum distance between any part of the top face of the base and a horizontal plane touching that face, and the maximum distance between any part of the top face of the lid and a horizontal plane touching that. The standard sets an acceptable limit of 1 for dishes below 100 mm in size and 1.5 for dishes above 100 mm in size. Finally, this standard describes a method for testing the fracture resistance. Using the apparatus described above, a steadily increasing force is applied with the test finger from 0 to 19.61 N in 10–15 sec. While the load is applied, the occurrence of cracks or other permanent deformities is observed, and the load level at which they occurred noted. The standard states that the lid should not fracture or permanently deform below 9.81 N, and the dish should fracture or permanently deform below 7.36 N.

ISO does not have a standard for plastic centrifuge tubes. Plastic centrifuge tubes are typically made of polyethylene terephthalate (PET), polypropylene, polycarbonate, or polystyrene [139]. Tubes need to be able to withstand high rotation speeds and forces, as much as 20,000 xg [139]. ISO 6427:2013 details the determination of extractables by organic solvents for plastics [140]. The testing apparatus in which the ground polymer samples are processed with solvents near their boiling point. The results are given as the ratio of the extracted mass to the original mass, expressed as a percentage. The National Standards Authority of Ireland (NSAI) has a standard S.R. CEN/TR 15932:2010 relating to bioplastics, which defines terminology [141], it prescribes definitions for organic material, polymer, plastic, renewable resource, biomass, biobased, biobased carbon content, biomass content, biocompatibile, biodegradable, biobased polymer, and biocomposite. NSAI standard I.S. EN 13432:2001 gives the biodegradability requirements for packaging, and I.S. EN 14995:2006 describes general evaluation of plastics compostability [142,143].

## 5. Environmental Impact of Laboratory Plastics

### 5.1. Plastic Labware End of Life

Most plastic labware is single-use, and used plastic lab consumables typically end up in landfill sites or incineration facilities. The recycling of plastic labware (or its composting, in the case of bioplastics), requires that the used consumables be washed and free of hazardous chemicals. Moreover, labware that had been in contact with living organisms needs to be sterilized before recycling or disposal.

If PLA is used in lab consumables, the necessary post-use treatment could have an effect on the material’s biodegradability and recyclability [73]. The presence of functional additives, used to improve the mechanical properties of labware PLA, may also have an impact on the recycling and composting processes. Scaffaro et al. [144] report a study on the recyclability of PLA enhanced with additives, stating that each recycling cycle caused changes in crystallinity that result in decreased molecular weight. If composting is selected for the labware’s end of life, eco-friendly additives should be used to ensure sustainability [101]. Lastly, the presence of additives may slow down the decomposition process [145].

#### End of Life for PLA

Used PLA can be recycled or disposed of in an environmentally-conscious manner using three main approaches: mechanical recycling, chemical recycling and composting. Mechanical recycling (consisting of washing the used PLA, grinding it and finally extruding recycled PLA) appears to have the lowest environmental impact from a life cycle assessment point of view [146,147]. Chemical recycling involves washing and grinding of the used PLA, followed by its hydrolysis to lactic acid, a concentration step and finally, polymerization and extrusion. This process has been shown to be more favourable than producing lactic acid from glucose fermentation, but the overall PLA recovery process requires more energy than mechanical recycling [147]. Composting, presumably the simplest way of disposing of used PLA, results in the highest overall environmental impact. It should be noted that PLA is only readily degradable in industrial settings and with the use of proper enzymes. With adequate conditions, ground PLA undergoes aerobic degradation producing organic matter and CO_2_. The environmental impact of composting is higher than that of chemical or mechanical recycling since no new PLA is made, thus requiring a manufacturing step to replace the composted plastic [146,148].

### 5.2. Carbon Dioxide Emissions of Plastic Labware Manufacture

Commercial scale production and application of synthetic plastics would enable the chance of entering the environment at a fast phase. Almost many investigations focused on the toxicity, comportment and fate with regard to environmental impact but failed to pay attention towards the effect of greenhouse gas (GHG) emissions and climate change. There is great concern over the increase the pollution over the concomitant increase in plastic wastes. It is evidenced that GHG emissions occur at all stage of plastic’s cycle that includes, extraction, transportation of plastic raw materials, manufacturing, waste recycling etc. The starting point of GHG emission begins from its extraction stage (raw materials) to the manufacture of plastics by oil and gas industries [149]. As per the report of Dormer et al. [150], carbon footprint of 1 kg of recycled PET trays (used in case of mushroom packaging) containing 85% recycled content was 1.538 kg CO_2_e. Hence, replacing conventional plastics with bio-plastics (made from renewable feedstocks like corn or switchgrass) is more often proposed strategy to mitigate the aforementioned environmental impacts. Renewable feedstock pathways viz. corn-based biopolymers produced with conventional energy are the dominant foreseen biopolymer option, and can reduce industry-wide GHG emissions by 25%, or 16 Mt CO_2_e/yr. It is also believed that in a long run, the manufacture of bio-polymers from advanced feedstocks coupled with renewable energy can ensure carbon neutral material production [151].

### 5.3. Environmental Considerations of PLA Feedstock

A consensus definition does not yet exist for a bio-based plastic (biopolymer). The US agriculture department defines a bio-based plastics as “a product that is partially or fully made from biological resources which includes material of agro- or forest residues” [152]. Sources of bio-based materials include all plant and animal mass derived from CO_2_ recently fixed via photosynthesis, per definition of a renewable resource. However, in actual, most of the bio-based plastics that exist on-date in market are blends of both bio- and petroleum-based materials. Fact is a bio-based plastic material is not necessarily sustainable; because there is a series of associated issues, right from the source material, production process, until its life cycle management. It is worth noting that producing PLA using a feedstock that competes with food production (e.g., corn, sugarcane etc.) is counterproductive to sustainability efforts. This is because such practices increase pressure on land use, which is already at capacity in Ireland due to the requirements of food production, biodiversity protection, and industrial and domestic construction. Considering all these facts the industrial production, research and commercialization of these materials are becoming very competitive and challenging. Thus, the commercialization of bio-based plastics is still infancy [153]. Hence, a way must be proposed by using a currently underutilized waste stream for PLA production which does not compete with food production and biodiversity protection for land, water, and energy use.

### 5.4. Economic Considerations of PLA Production

Production costs for bioplastics is largely defined by production scale and costs of raw materials. Typically, production costs of bioplastics are 3 to 4 times higher than petroleum-based alternatives [154]. Techno-economic analysis reports have shown that production costs for PLA can vary widely with scale, from a peak of € 3.56/kg using a small capacity plant producing 10,624 t/yr to € 0.91/kg for a larger production facility offering a manufacturing capacity of 100,000 t/yr [155]. PHA as an alternative bioplastic has comparable production costs ranging between € 1.1 per kg [156] to € 5.24 per kg [157] (using an exchange rate of 1 USD = 0.9138 EUR on 15 March 2022). It has been reported that the minimum sale price of PLA is € 3/kg [158], indicating that large-scale production of 100,000 t/yr is required to produce economically viable biopolymers. Given that the global production capacity of PLA worldwide is currently estimated at 0.46 Mt/yr (2021) and expected to grow to 0.79 Mt/yr by 2026 [50], economies of scale will ensure that PLA production is economically viable in years to come.

## 6. Conclusions

The advantageous properties of plastics have made them ubiquitous materials. However, they have two main sustainability flaws; dependence on non-renewable, ecologically unfriendly fossil fuels as a source, and the generation of non-degradable waste. The accumulation of this waste and its penetration into land, river, and sea ecosystems is a serious environmental concern. This has driven interest in the production of sustainable bioplastics. These plastics may be generated from biological waste sources, reduce dependence on fossil fuels, and be biodegradable. Single-use plastic laboratory consumables alone have been estimated to generate 5.5 Mt in one year. If these single-use items could be manufactured using a bioplastic like PLA, it could be a significant benefit to the environment and sustainability. In this report, the properties of common labware polymers, and PLA, have been reviewed, along with the standards for plastic labware. PLA’s properties have some limitations in terms of brittleness and temperature & solvent resistance; however, the material’s properties can be tuned by the blend of isomers that are polymerised to make the plastic, the level of crystallinity, and through the addition of plasticisers (such as the monomer lactic acid itself) and other additives. Some commercial labware items are produced with PLA, and some research into additive manufacturing of PLA labware has been undertaken, but uptake of bioplastics for labware is low. More research is needed to establish for which forms of PLA or PLA-based composites can be suitable for labware, and which labware items they may be suitable for.

## Figures and Tables

**Figure 1 materials-15-02989-f001:**
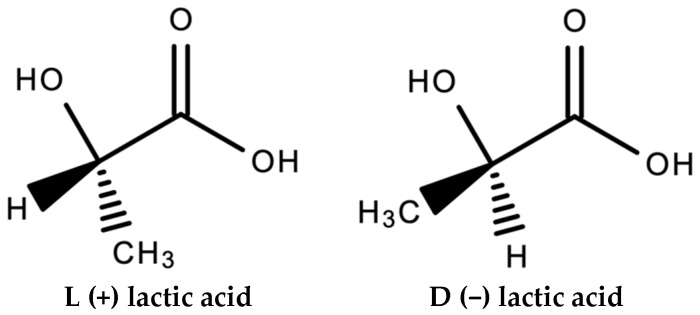
Stereoisomers of lactic acid.

**Figure 2 materials-15-02989-f002:**
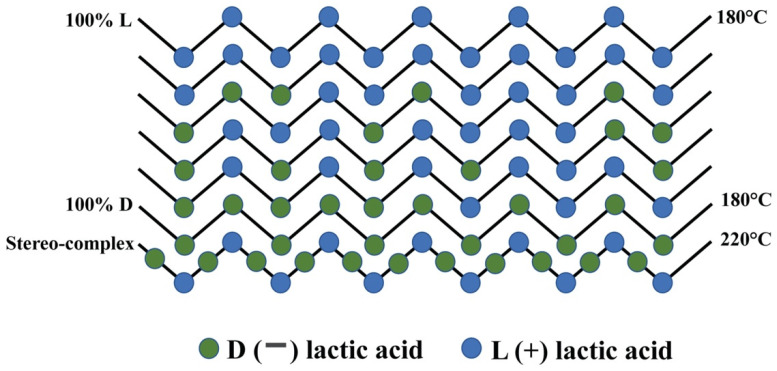
Examples of molecular configurations of PLA obtained through combining the two lactic acids.

**Figure 3 materials-15-02989-f003:**
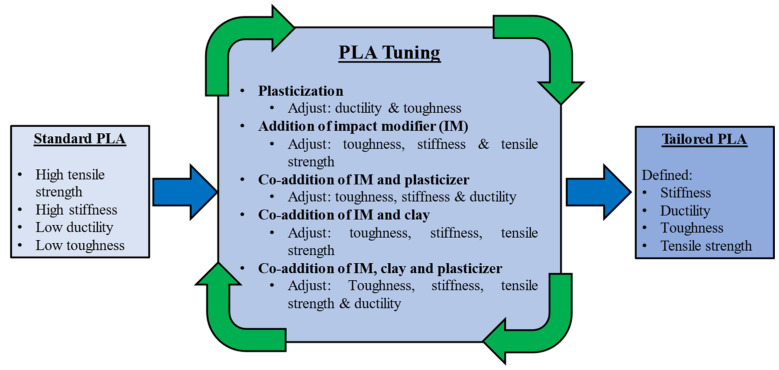
Tuning the properties of PLA with a variety of additives.

**Figure 4 materials-15-02989-f004:**
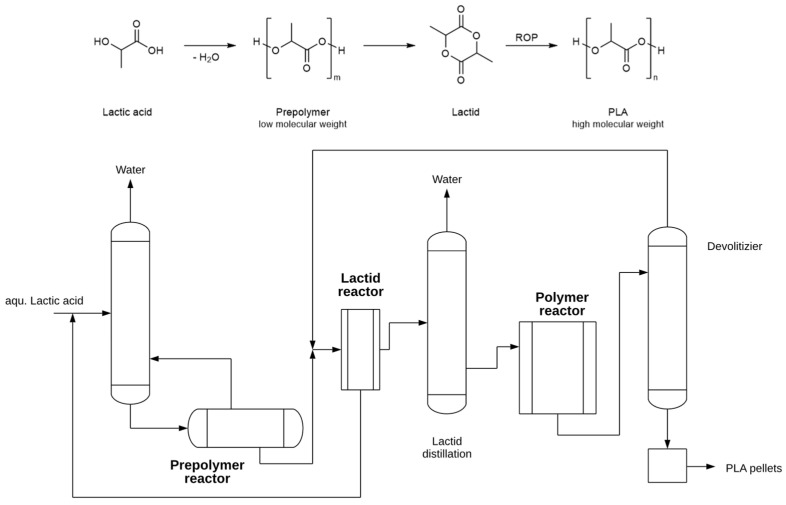
Schematic diagram of a typical combined lactide/ROP process for production of PLA.

**Table 1 materials-15-02989-t001:** Summary of common lab ware plastics, their typical items, and their general properties.

Polymer	Typical Items	General Properties
Polymethyl pentene (PMP)	Beakers, Cylinders, Erlenmeyer Flasks, Jars	Rigid, translucent, fair UV resistance
High Density Polyethylene (HDPE)	Bottles, Carboys, Pans	Semi-rigid, translucent, poor UV resistance
Low Density Polyethylene (LDPE)	Bottles, Carboys, Wash or Dropper Bottles	Flexible, translucent, fair UV resistance
Polypropylene (PP)	Autoclave baskets, Carboys, Funnels, Vacuum Flasks	Rigid, translucent, fair UV resistance
Polypropylene Copolymer (PPCO)	Bottles, Beakers, Centrifuge Tubes, Graduated Cylinders	Semi-rigid, translucent, fair UV resistance
Polyvinyl Chloride (PVC)	Tubing	Very flexible, transparent
Polyethylene Terephthalate G Copolyester (PETG)	Bioprocessing Containers, Bottles, Erlenmeyer Flasks	Moderately flexible, transparent, fair UV resistance
Polytetrafluoroethylene (PTFE)	Stirrers, Test Tubes, Vessels	High thermal stability and chemical inertness
Polystyrene (PS)	Filtration Units, Pipettes/Tips, Single-Use Petri Dishes	Rigid, transparent, fair chemical resistance, poor UV resistance
Polycarbonate (PC)	Bottles, Culture Flasks, Desiccators, Jars	Rigid, transparent, fair UV resistance
Polysulfone (PSF)	Bottles, Centrifuge Tubes, Filtration Units	Rigid, transparent, poor UV resistance
Teflon (FEP)	Bottles, Centrifuge Tubes, Wash bottles	Very flexible, translucent, good UV resistance
Teflon (PFA)	Beakers, Bottles, Cylinders, Tubing	Very flexible, translucent, fair UV resistance

**Table 2 materials-15-02989-t002:** Properties of general purpose, medium impact, and high impact polystyrene [15,16,17].

Properties	General Purpose
Specific Gravity	1.04
Specific Heat (J/kg K)	1256–1465
Thermal Conductivity (W/mK)	0.100–0.156
Thermal Expansion (K^−1^)	5.94–8.64 × 10^−5^
Ultimate Tensile Strength (MPa)	34.5–68.9
Yield Strength (MPa)	34.5–68.9
Flexural Strength (MPa)	68.9–103
Impact Strength-Izod notched (J/m)	-
Tensile Elastic Modulus (GPa)	3.17–3.45
Flexural Elastic Modulus (GPa)	2.76–3.45
Yield Elongation (%)	1–2.3
Max Elongation (%)	1.0–2.3
Hardness (Rockwell)	M72
Refractive Index	1.6
Water Absorption (% in 24 h)	0.03–0.2

**Table 3 materials-15-02989-t003:** Properties of PET and PETG [21,22,23].

Properties	PET	PETG
Specific Gravity	1.38	1.27
Thermal Conductivity (W/mK)	0.29	-
Glass Transition Temperature	340–413	354
Thermal Expansion (m/mK)	6.84	6.84
Tensile Strength (MPa)	58.6–72.4	53.1
Flexural Strength (MPa)	96.5–124.1	77.2
Impact Strength-Izod notched (J/m)	13.34–34.68	90.8
Tensile Elastic Modulus (GPa)	2.7–4.1	2.21
Flexural Elastic Modulus (GPa)	2.4–3.1	2.14
Max Elongation (%)	30–80	-
Hardness (Rockwell)	M50–100	115 (R Scale)
Refractive Index	1.58	1.57
Water Absorption (% in 24 h)	0.1–0.2	0.2

**Table 4 materials-15-02989-t004:** General properties of Polycarbonate [30,31,32].

Properties	Polycarbonate
Specific Gravity	1.2
Specific Heat (J/kg K)	1260
Thermal Conductivity (W/mK)	0.190
Thermal Expansion (K^−1^)	6.75 × 10^−6^
Ultimate Tensile Strength (MPa)	65.5
Yield Strength (MPa)	58.6
Flexural Strength (MPa)	93.1
Impact Strength-Izod notched (J/m)	641–854
Tensile Elastic Modulus (GPa)	2.38
Flexural Elastic Modulus (GPa)	2.34
Yield Elongation (%)	5
Max Elongation (%)	110
Hardness (Rockwell)	M70
Refractive Index	1.586
Water Absorption (% in 24 h)	0.15

**Table 5 materials-15-02989-t005:** General properties of Polypropylene [37,38].

Properties	General Purpose
Specific Gravity	0.90–0.91
Specific Heat (J/kg K)	1880
Thermal Conductivity (W/mK)	2.09–2.35
Thermal Expansion (K^−1^)	6.84–10.44 × 10^−5^
Tensile Strength (MPa)	31.0–41.4
Yield Strength (MPa)	31.0–41.4
Flexural Strength (MPa)	41.4–48.3
Impact Strength-Izod notched (J/m)	21.4–117
Tensile Elastic Modulus (GPa)	-
Flexural Elastic Modulus (GPa)	1.17–1.72
Yield Elongation (%)	9–15
Max Elongation (%)	100–600
Hardness (Rockwell)	R80–100
Refractive Index	Opaque
Water Absorption (% in 24 h)	<0.01–0.03

**Table 6 materials-15-02989-t006:** General properties for PLA [54,55].

Properties	PLA
Specific Gravity	1.24
Tensile Strength (MPa)	62.1
Tensile Elongation (%)	3.5
Impact Strength-Izod notched (J/m)	16
Flexural Strength (MPa)	108
Flexural Modulus (MPa)	3600
Glass Transition Temperature (K)	328
Melting Temperature (K)	428
Heat Distortion Temperature (K)	328
Clarity	Transparent

**Table 7 materials-15-02989-t007:** Properties of PLA, PS, and PET [15,16,17,21,23,54,57].

Properties	PLA	PS	PET
Density (kg/m^3^)	1.26	1.05	1.40
Ultimate Tensile Strength (MPa)	59	45	57
Elastic Modulus (GPa)	3.8	3.2	2.8–4.1
Max Elongation (%)	4–7	3	300
Impact Strength-Izod notched (J/m)	26	21	59
Heat Deflection (°C)	55	75	67

**Table 8 materials-15-02989-t008:** Mechanical properties of PLA variants with varying molecular weights [58].

**PLLA**				
**Molecular Weight (g/mol)**	**23,000**	**31,000**	**58,000**	**67,000**
Ultimate Tensile Strength (MPa)	59	55	58	59
Yield Strength (MPa)		65	68	70
Max Elongation (%)	1.5	5.5	5.0	7.0
Yield Elongation (%)		2.2	2.3	2.2
Tensile Elastic Modulus (GPa)	3.55	3.55	3.75	4.75
Flexural Strength (MPa)	64	97	100	106
Max Flexural Strain (%)	2.0	4.2	4.1	4.7
Flexural Elastic Modulus (GPa)	3.65	3.60	3.60	3.65
Impact Strength-notched (J/m)	19	22	25	26
Impact Strength-unnotched (J/m)	135	175	185	195
Heat Deflection Temperature (°C)	57	55		55
Vicat Penetration (°C)	60	59	59	59
Rockwell Hardness (HR)	85	84	83	88
**Annealed PLLA**				
**Molecular Weight (g/mol)**	**20,000**	**33,500**	**47,000**	**71,000**
Ultimate Tensile Strength (MPa)	47	54	59	66
Yield Strength (MPa)		63	68	70
Max Elongation (%)	1.3	3.3	3.5	4.0
Yield Elongation (%)		1.8	2.2	2
Tensile Elastic Modulus (GPa)	4.10	4.10	4.05	4.15
Flexural Strength (MPa)	51	83	113	119
Max Flexural Strain (%)	1.6	2.3	4.8	4.6
Flexural Elastic Modulus (GPa)	4.20	4.00	4.05	4.15
Impact Strength-notched (J/m)	32	55	70	66
Impact Strength-unnotched (J/m)	180	360	340	350
Heat Deflection Temperature (°C)	66	60		61
Vicat Penetration (°C)	157	159	163	165
Rockwell Hardness (HR)	84	82	84	88
**PDLLA**				
**Molecular Weight (g/mol)**	**47,500**	**75,000**		**114,000**
Ultimate Tensile Strength (MPa)	40	44		44
Yield Strength (MPa)	49	53		53
Max Elongation (%)	7.5	4.8		5.4
Yield Elongation (%)	1.7	1.4		1.5
Tensile Elastic Modulus (GPa)	3.65	4.05		3.90
Flexural Strength (MPa)	84	86		88
Max Flexural Strain (%)	4.8	4.1		4.2
Flexural Elastic Modulus (GPa)	3.50	3.55		3.60
Impact Strength-notched (J/m)	18	17		18
Impact Strength-unnotched (J/m)	135	140		150
Heat Deflection Temperature (°C)	51	50		50
Vicat Penetration (°C)	52	53		52
Rockwell Hardness (HR)	78	72		76

**Table 9 materials-15-02989-t009:** Properties of PLA with different isomer types and varying molecular weights [59].

Isomer Type	*M_n_*	*M_w_/M_n_*	*T_g_*	*T_m_*	*ΔH_m_*	*T_c_*	*ΔH_c_*
L	4700	1.09	45.6	157.8	55.5	98.3	47.8
DL	4300	1.9	44.7	-	-	-	-
L	7000	1.09	67.9	159.9	58.8	108.3	48.3
DL	7300	1.16	44.1	-	-	-	-
D	13,800	1.19	65.7	170.3	67.0	107.6	52.4
L	14,000	1.12	66.8	173.3	61.0	110.3	48.1
D	16,500	1.2	69.1	173.5	64.6	109.0	51.6
L	16,800	1.32	58.6	173.4	61.4	105.0	38.1

*M_n_*—Number average molecular weight (g/mol). *M_w_/M_n_*—Dispersity index. *T_g_*—Glass transition temperature (°C). *T_m_*—Melting temperature (°C). *ΔH_m_*—Melting enthalpy (J/g). *T_c_*—Crystallisation temperature (°C). *ΔH_c_*—Crystallisation enthalpy (J/g).

**Table 10 materials-15-02989-t010:** Sterilisation techniques and their advantages and disadvantages for use with PLA.

Techniques	Conditions	Advantages	Disadvantages
Steam	High steam pressure, 120–135 °C	No toxic residue	Deformation or degradation due to water attack, limited usage for lactic acid-based polymers
Dry heat	160–190 °C	No toxic residue	Melting and softening of polymer, not usable for lactic acid-based polymers
Radiation	Ionising or gamma	High penetration, low chemical reactivity, and quick effect	Instability and deterioration, crosslinking or breaking of polymer chains
Gas	Ethylene oxide	Low temperature range	Lengthy process due to degassing, residues are toxic

**Table 11 materials-15-02989-t011:** Solubility parameters of key solvents at 25 °C [75,77,78].

Solvents	*δ_d_*	*δ_p_*	*δ_h_*	*δ_t_*
Acetone	15.0	10.4	7.0	19.6
Acetonitrile	15.3	18.0	6.1	24.4
Benzene	18.4	0.0	2.0	18.5
Chloroform	17.8	3.1	5.5	18.9
*m*-Cresol	18.0	5.1	12.9	22.7
Dimethyl formamide	17.4	13.7	11.3	24.9
Dimethyl sulfoxide	18.4	16.4	10.0	26.6
1-4 Dioxane	19.0	1.8	7.4	20.5
1-3 Dioxolane	18.1	6.6	9.3	21.4
Ethyl acetate	15.8	5.3	7.2	18.2
Furan	17.8	1.8	5.3	18.7
Hexafluoro isopropanol	17.2	4.5	14.7	23.1
Isoamyl alcohol	15.8	5.2	13.3	21.3
Methylene dichloride	18.2	6.3	6.1	20.2
Methyl ethyl ketone	16.0	9.0	5.1	19.1
*N*-Methyl pyrrolidone	18.0	12.3	7.2	23.0
Pyridine	19.0	8.8	5.9	31.8
Tetrahydrofuran	16.8	5.7	8.0	19.5
Toluene	18.0	1.4	2.0	18.2
Xylene	17.6	1.0	3.1	17.9
Isopropyl ether	13.7	3.9	2.3	14.4
Cyclohexane	16.5	0.0	0.2	16.5
Hexane	14.9	0.0	0.0	14.9
Ethanol	15.8	8.8	19.4	26.5
Methanol	15.1	12.3	22.3	29.6
Water	15.5	16.0	42.3	47.8
Diethyl ether	14.5	2.9	5.1	15.6

*δ_d_*—Dispersion solubility parameter. *δ_p_*—Polar solubility parameter. *δ_h_*—Hydrogen solubility parameter. *δ_t_*—Total solubility parameter. The SI units of the above solubility parameters are MPa^0.5^.

**Table 12 materials-15-02989-t012:** Solubility parameters for PLA using an appropriate method at 25 °C [75].

Method	*δ_d_*	*δ_p_*	*δ_h_*	*δ_t_*
Intrinsic 3D viscosity method	17.61	5.30	5.80	19.28
Intrinsic 1D viscosity method	-	-	-	19.16
Classical 3D geometric method	16.85	9.00	4.05	19.53
Fedors group contribution	-	-	-	21.42
Van Krevelen group contribution	-	-	-	17.64
Optimisation method	18.50	9.70	6.00	21.73

**Table 13 materials-15-02989-t013:** ISO standards relating to plastic laboratory wares.

ISO No.	ISO Name
384:2015	Laboratory glass and plastics ware—Principles of design and construction of volumetric instruments
6706:1981	Plastics Laboratory Ware-Graduated Measuring Cylinders
7056:1981	Plastics Laboratory Ware-Beakers
7057:1981	Plastics Laboratory Ware-Filter Funnels
12771:1997	Plastics Laboratory Ware-Disposable Serological Pipettes
24998:2008	Plastics Laboratory Ware-Single-Use Petri Dishes for Microbiological Procedures

## Data Availability

Not applicable.

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
