# Peer review of "A Review of Polylactic Acid as a Replacement Material for Single-Use Laboratory Components"

_materials, 2022, doi:10.3390/ma15092989_

Round 1

Reviewer 1 Report

The topic is of a wider interest. No review can cover everything; however the choice of materials and properties referred in the article looks sometimes chaotic, as mentioned below in specific comments. When summarizing specific properties of materials in tables, involving data from more sources with proper citations is expected from a review article. The authors do not respect standards of writing variables and physical quantities, tables are not clearly arranged. A random check has discovered a piece of information coming from different source than indicated in the text.

Table 1 is not comfortable for reading since table rows consist of more than 1 line of text and they are not optically divided each from other. Please, try to divide the table rows e.g. using thin dotted lines. The "References" columns looks strange when it contains values for the 7th row only (the same for other tables). References should be given for all items, and it seems that they should have been stated rather in the table legend than in the separate column if they are common for the whole table.

Many tables present the values from one, two, or three sources. In a review article, I would expect tables getting together information from more sources, indicating the source for each item.

In 2.1 section, the properties of only some of materials listed in Table 1 are given in detail. At least a paragraph mentioning the reason for selecting four groups of polymers to be described in more details and commenting the remaining materials should be included.

Line 73: The 2.1.2 section heading Polyethylene is misleading, since its contents deals with polyesters (PET,PETG).

The text before Table 8 leads to an expectation of comparison of PLLA and PDLLA. PDLLA is not mentioned in the table, while the third group is not labelled. Is the third group PDLLA?

Line 206, 210, 212 etc.: If authors are mentioned, I would expect the reference immediately after names.

Line 293:?d is repeated, while ?t is proper for this line.

Table 13: Explanation of exact presented numbers meaning is missing.

Line 403: The reference [99] is given, but the information in the begin of the paragraph has not been found in this ref. I have found it in ref. [98].

Line 428: The comma after names is confusing, the sentence looks like an appeal to them to note the advantages. Similar extra commas misleading at least for non-English speaker occur here and there in the whole text. In line 441, space missing after fullstop.

Line 462 - table 15: Symbol explanation relating not to specific table cell, but to the meaning of the symbol in the whole table, should be given in the legend, not using superscript letters at the first occurrence.

Line 581: "approximately 0" is unacceptably unclear quantification. It should be used at least together with units - 0 tonnes, 0 kg, 0 million tonnes, but the expected order of emissions amount should be specified (e.g. thousands tons  per year).

Quantities and units are not always properly used, in particular as to used upright and italics.
The IUPAC recommended documents are https://iupac.org/wp-content/uploads/2019/05/IUPAC-GB3-2012-2ndPrinting-PDFsearchable.pdf 
and https://iupac.org/wp-content/uploads/2016/01/ICTNS-On-the-use-of-italic-and-roman-fonts-for-symbols-in-scientific-text.pdf
The IUPAP recommendation is available for example at https://iupap.org/wp-content/uploads/2021/03/A4.pdf. 
In the paper being reviewed, the authors use upright font for physical quantities, although the IUPAC and IUPAP recommended style is in italics.
After possible correction, checking the used fonts during proofreading can be recommended - the experience with MDPI journals is that correctly used symbols in the submitted or accepted manuscripts tend to be changed to wrong during production.

When comparing the references list with published paper with related contents, I have not found within the references the following:
 https://doi.org/10.1002/pen.25623
 https://doi.org/10.1016/j.eurpolymj.2007.06.045
 https://doi.org/10.3390/polym11071193 
 https://doi.org/10.22141/2224-1507.9.1.2019.163056
 https://doi.org/10.1016/S0141-3910(02)00372-5
 https://doi.org/10.1007/s10439-015-1455-8
 https://doi.org/10.1016/j.jmbbm.2019.103510
 https://doi.org/10.1016/j.matdes.2017.11.031
 https://doi.org/10.1016/j.addma.2020.101414
 https://doi.org/10.1002/app.47824
 https://doi.org/10.1002/masy.201900064
 https://doi.org/10.3390/polym11091496
 https://doi.org/10.1021/acs.iecr.7b00930
 https://doi.org/10.3390/su12020652
 https://doi.org/10.3390/app10020509
 https://doi.org/10.3390/polym11121908
 https://doi.org/10.14314/polimery.2020.7.8 
 https://doi.org/10.1016/j.compositesb.2018.04.017
 https://doi.org/10.1002/pi.5588
 https://doi.org/10.1007/s00068-020-01564-1

I do not insist on that they are relevant references, but the authors can comment in the reply why these papers are not relevant.

I declare that I am not an author or coauthor of any of the suggested references, I do not know personally neither have ever in past communicated with authors of any of mentioned papers, nor have any connection with them. I am not interested in citations of  any of the listed papers.

Author Response

Thank you very much for reviewing our manuscript, we appreciate you comments, and suggestions to improve the review article. Please see a list of the changes made below. Thank you.

  1. We have tidied up and rearranged the tables, removed unnecessary subscripts, and performed a check on the text. The choice of material covered was directed toward typical labware components.
  2. Table 1 was rearranged, and dotted lines used as internal borders. References were placed along the Table tittle and removed from all tables in the text.
  3. The most suitable sources for data for each material property was used in the review, some individual sources provided sufficient information to describe the required material properties table.
  4. In this review the authors utilised 131 sources, the authors felt this was enough to offer a substantial review of the field, however if the reviewer requires, the authors will be happy to add the additional sources mentioned in the comments.
  5. Line 73: The 2.1.2 section heading Polyethylene is misleading, since its contents deals with polyesters (PET,PETG). – it was changed to the following: “2.1.2. Polyethylene Terephthalate G Copolyester.”
  6. The text in table 8 was amended, adding “PDLLA”.
  7. Line 206, 210, 212 etc.: Amended in the text so that reference is immediately after names.
  8. Line 293:?d is repeated, while ?t is proper for this line. – it was changed in text so that ? t is used: 4 ? t –Total solubility parameter.
  9. Table 13: Units added to the temperature column.
  10. Reference [99] in line 401 was changed to reference [99].
  11. The comma was removed from line 428 and 441.
  12. Line 462, subscripts were removed for clarity.
  13. Line 581 was reworded as follows: “the manufacture of bio-polymers from advanced feedstocks coupled with renewable energy can ensure carbon neutral material production”.
  14. In section 2.1, the following reasoning was added: The most common polymers used for single-use labware will be discussed in more detail in the review including: PETG, PET, PC, PP, PS and PLA.

Reviewer 2 Report

In the manuscript, the authors present the scale of the ecological problem that is generated by the currently used level of disposable laboratory equipment made of classic plastics. They indicate that a significant benefit for the environment and sustainable development would be the use of bioplastics produced from renewable raw materials in the production of this type of object. In the manuscript, the authors have limited to the possibility of using PLA  in the production of this type of equipment.

In the summary, the authors  indicate the current low use of PLA for the production of disposable laboratory equipment and the need to modify the properties of PLA towards obtaining polymeric materials with properties that meet the standards specified for polymer materials for the  laboratory equipment production

Specific comments:

Due to the fact that It is difficult to replace the various polymeric materials used so far with one PLA polyester. The work would be more interesting if, in addition to PLA, the authors put attention to the use in this area of other biodegradable polymers e.g  family, of aliphatic polyesters PHAs obtained via  biotechnological way, also with the use of waste materials.

Additionally, an interesting and important element of the manuscript would be to look at the application of these bioplastics from an economic point of view

There is a mistake in the description of Table 9

"Table 9. Properties of general purpose, medium impact, and high impact polystyrene"

Author Response

The authors would like to thank the reviewer for the valuable comments. Please see our response to your comments below.

  1.  Text was added discussing briefly PHAs:  Polylactic Acid (PLA) is one of the most commonly used bioplastics, in 2021 it was reported to hold the largest market share worldwide [60] for biodegradable bioplastics manufacturing capacity. PLA‘s manufacturing capacity accounted for 42% of the total biodegradable, bioplastics production worldwide, with its nearest rival polyhydroxyalkanoates (PHA) [7], [61] accounting for only 4% of worldwide production [60].
  2. Also, the authors would argue that given PLAs huge market share and prevalence a dedicated review examining the material alone and its application as a replacement material is justified.
  3. Text was added to discuss the economic conditions of PLA production: Production costs for bioplastics is largely defined by production scale and costs of raw materials. Typically, production costs of bioplastics are 3 to 4 times higher than petroleum-based alternatives [5]. Techno-economic analysis reports have shown that production costs for PLA can vary widely with scale, from a peak of €3.56/Kg using a small capacity plant producing 10,624 tonnes/year to €0.91/Kg for a larger production facility offering a manufacturing capacity of 100,000 tonnes/year [3]. PHA as an alternative bioplastic has comparable production costs ranging between 1.1 €/Kg [2] to 5.24 euro/kg [1] (using an exchange rate of 1 USD = 0.9138 EUR on 15th March 2022). It has been reported that the minimum sale price of PLA is €3/Kg [4], indicating that large-scale production of 100,000 tonnes/year is required to produce economically viable biopolymers. Given that the global production capacity of PLA worldwide is currently estimated at 0.46 million tonnes/year (2021) [60] and expected to grow to 0.79 million tonnes/year by 2026 [60], economies of scale will ensure that PLA production is economically viable in years to come.
  4. The mistake in Table 9 was fixed as follows: “Table 9. Properties of PLA with different isomer types and varying molecular weights.”

Thank you for your feedback, we hope this addresses your comments.

Reviewer 3 Report

The manuscript titled "A review of Polylactic acid as a replacement material for single-use laboratory components" by Freeland et al. is a comprehensive literature review that suggests the use of PLA as a substitute for petroleum-based plastics in manufacturing single-use laboratory components. The manuscript is well organized and covers a much-needed discussion on the aforementioned topic. I recommend this article for publication in Materials after the following minor changes:

  1. The title of section 2.1.2 is Polyethylene, which seems to be misleading. The description is about PET and PETG. Please check whether it is correct or should be changed.
  2.  In Table 4, change PET with Polycarbonate.
  3.  In Table 6, change PET with PLA.
  4.  In Table 8, the name of the third PLA variant is not mentioned. Please mention it.
  5. In table 9, I don't think there is a requirement of mentioning superscripts a,b,c,d,e,f, etc. The authors can remove the superscripts and only use the abbreviated forms to give their descriptions in the table footer.

Author Response

Thank you for your comments, the following changes have been made after your recommendations:

  1. The title of section 2.1.2  has been changed in the text to: “Polyethylene Terephthalate G Copolyester“
  2.  In Table 4, "PET" has been changed to read "Polycarbonate" highlighted in yellow.
  3.  Table 6, "PET" has been changed to  "PLA".
  4.  Table 8, the name of the third PLA variant is now mentioned, Titled:“PDLLA“.
  5. In table 9, subscripts removed.

Reviewer 4 Report

After review the manuscript, I consider that work do not make a significant contribution, due the content is not resolutive and it is not related completely with title, due the title indicate the PLA as a replacement material for laboratory components, there is not potential applications of PLA for laboratory tools, only describes some other polymers that are used for this applications. Other specific comments are detailed following:

-The section 2.1.2 title is Polyethylene, but text is referred to Polyethylene terephthalate G copolymer, those materials are not the same, this is a significant mistake. 

-Why a comparison of properties among the described polymers that usually are used for laboratory accesories is not done?

-In section 2.2 describe the production techniques for plastic lab ware, bu which of those techniques apply for PLA?

-In line 187 indicate that table 9 show the results for varied PLA samples, but table caption indicate that those data corresponds to HIPS properties, and also the text is confuse about which is the references 50 or 51.

-table 9 bottom, describe the symbols used in table, but Mw/Mn is not molecular weight, it corresponds to polydispersity index.

-In section 4.1 describes the standard for plastic labotary wares, but did not indicate if PLA can fix in each of them, which will be the interesting contribution.

Author Response

Thank you for your detailed comments and suggestions for improvements for our review paper. We have addressed your comments below in bold, thank you for your input.

After review the manuscript, I consider that work do not make a significant contribution, due the content is not resolutive and it is not related completely with title, due the title indicate the PLA as a replacement material for laboratory components, there is not potential applications of PLA for laboratory tools, only describes some other polymers that are used for this applications.  - PLA-based lab ware components is a newly developing field, currently there are only a few applications of PLA-based lab ware described in Section 3.9.1. PLA for labware applications. This review paper aims to identify the potential for PLA-based lab ware, discussing its mechanical, thermal, optical, chemical and biological properties, in terms of labware requirements.

Other specific comments are detailed following:

-The section 2.1.2 title is Polyethylene, but text is referred to Polyethylene terephthalate G copolymer, those materials are not the same, this is a significant mistake.  - The title has been amended to read: “Polyethylene terephthalate G copolymer” in the text.

-Why a comparison of properties among the described polymers that usually are used for laboratory accesories is not done? - The authors aim to discuss the most prominently used labware plastics in the text.

-In section 2.2 describe the production techniques for plastic lab ware, but which of those techniques apply for PLA? Text added: "The most common techniques employed are 3D printing, injection moulding and blow moulding, which can also be applied to PLA-based components. "

-In line 187 indicate that table 9 show the results for varied PLA samples, but table caption indicate that those data corresponds to HIPS properties, and also the text is confuse about which is the references 50 or 51. - corrected in the text to the following: “Table 9: Properties of PLA with different isomer types and varying molecular weights. “

-table 9 bottom, describe the symbols used in table, but Mw/Mn is not molecular weight, it corresponds to polydispersity index. Changed in text to “polydispersity index”.

-In section 4.1 describes the standard for plastic laboratory wares, but did not indicate if PLA can fix in each of them, which will be the interesting contribution. Text added: “To-date no literature has directly indicated if PLA can meet the ISO requirements described below. However, the standards remain the framework that PLA-based components must meet to be adopted as plastic lab-ware alternatives. The material properties of PLA described in Table 8, along with functionality improvements using additives illustrated in Table 9 suggest that targeted PLA-based labware components could be manufactured to meet a number of the ISO standards. “

Round 2

Reviewer 1 Report

I feel that the main weak point of the article is lack of the carefulness. One of the most striking demerits of this manuscript version is that IUPAC and IUPAP recommendations for writing physical quantities are still not respected. When some type of mistake was pointed out, authors did not check and remove it through the whole article.

line 61: As the reaction to the criticism of seemingly random choice of reported materials, the authors added: The most common polymers used for single-use labware will be discussed in more detail in the review including: PETG, PET, PC, PP, PS and PLA. Since PLA is not dealt in 2.1 together with other materials, but in 3, its mention in this form is misleading. The new sentence commenting detailed dealing with this polymer shouls be added.

line 73: The properties ... is shown is stated. I would expect The properties ... are shown. However, the authors are native English speakers and they surely know better than I what is correct in English.

line 76 Table 2: J/Kg/°C:  Using °C in the denominator does not reflect the nature of this unit. In addition lowercase k is IUPAP-recommended symbol for kilo-. Therefore, J/kg/K (or J kg-1 K-1) would be not by formally wrong. The same applies for Tables 3, 4, 5.

lines 155-159: Refs. [60] and [61] are used before [44] etc. References should have been renumbered.

line 197: Tm and Tg should be corrected to Tm and Tg (quantity in italics, subscripts upright)

line 198: Mn should be corrected to Mn

line 199: Tg should be corrected to  Tg

line 204-211 Table 9 and following lines: Mn, Mw/Mn, Tg, Tm, ΔHm, Tc, ΔHc should be corrected to Mn, Mw/Mn, Tg, Tm, ΔHm, Tc, ΔHc 

line 299: abbreviations for ortho-, meta-, para- should be in italics, change m-Cresol to m-Cresol. The n-Methyl pyrrolidone is improbable (n- prefix means a linear chain, while N-Methyl pyrrolidone is a common compound (see Letter symbols for elements are italic when they are locants in chemical-compound names indicating attachments to heteroatoms, e.g. O-, N-, S-, and P- in the document referred in the first review round). However, also an opposite recommendation can be found in case of heteroatoms; capital and not lowercase N is anyway most probably what should be here.

lines 299-303: I suggest to remove superscripts 1, 2, 3, 4; they look like solubility parameters powered to those values. It is misleading. The explanation without them is clear.

line 299 Table 11: Right (decimal) alignement would be more reader-friendly.

line 320 Table 13: Adding of the temperature units is appreciated. However, the meaning of the other numbers has to be searched in the text around. The table legend or header should tell the reader what the table presents. The Effect is not a physical quantity. The values are amounts of the migrated components in water, probably divided by the sheet area. This (or, if I understood wrongly, their correct meaning) should be stated in the heading and the legend. In addition, the sheet thinkness is to be indicated here. The currently presented values are too uncertain numbers. In addition, a and b superscripts are misleading, they should be removed.

line 326: The reference would be better immediately after the name of reported author. Similarly in lines 351, 353, 373, 379, 393, 464, 466, 557.

line 388, 389: Insert space between number and units.

line 396: Change TiO2 to TiO2 here and through the whole text.

line 432 Fig 4: If both reaction and aparature diagram are given in the figure, the reaction relating to each reactor should be graphically assigned. It seems to be space enough to increase the descriptions of apparature components to make them better readable.

line 438: The misleading comma persists, although its removal was reported.

line 451: Space after fullstop still missing.

line 472: Can you make the table wider and unwrap the rows?

line 600: Where is the end of double-quoted text?

Authors have added several references in the least laborius way, so that they have added a new paragraph just before conclusion and they need not renumber any references. Adding any missing information in the review article is welcome, however I feel that authors prefer to satisfy formally requierements of the reviewer rather than to rectify the demerits consisting in the too narrow selection of reported literature. This selection is appropriate for an original research article with the own results of authors. However, the review article without new results should, in my opinion, cover existing literature in a larger extent.

Ref. 99 is now in the reference list, but not used in the text. Please check what you have used from this paper and place it again.

I note also that some of other reviewers raised relevant and more serious objections than mine. Possible acceptation comes into question after fixing the demerits mentioned by others and after removing the formal mistakes listed in this report.

Author Response

Dear Reviewer,

Thank you very much for your comments and taking the time to review our article. We have tried to address them, please see our responses in bold below and in the uploaded article.

Kind Regards,

Brian Freeland

  1. line 61: As the reaction to the criticism of seemingly random choice of reported materials, the authors added: The most common polymers used for single-use labware will be discussed in more detail in the review including: PETG, PET, PC, PP, PS and PLA. Since PLA is not dealt in 2.1 together with other materials, but in 3, its mention in this form is misleading. The new sentence commenting detailed dealing with this polymer shouls be added. - PLA and PET is removed in the aforementioned highlighted line. (But they expect for writing with respect to other polymer too..!) Line no. 61-62 amended.
  2. line 73: The properties ... is shown is stated. I would expect The properties ... are shown. However, the authors are native English speakers and they surely know better than I what is correct in English. – changes made to line no. 73
  3. line 76 Table 2: J/Kg/°C:  Using °C in the denominator does not reflect the nature of this unit. In addition lowercase k is IUPAP-recommended symbol for kilo-. Therefore, J/kg/K (or J kg-1 K-1) would be not by formally wrong. The same applies for Tables 3, 4, 5. -Corrected with J/kg/K and Highlighted - In Table 2, 3, 4, 5
  4. lines 155-159: Refs. [60] and [61] are used before [44] etc. References should have been renumbered. - Corrected and Highlighted - Line No. 160-164
  5. line 197: Tm and Tg should be corrected to Tm and Tg (quantity in italics, subscripts upright)  - Corrected and Highlighted) - Line No. 206
  6. line 198: Mn should be corrected to Mn  - Corrected and Highlighted - Line No. 207
  7. line 199: Tg should be corrected to  Tg    - Corrected and Highlighted -  Line No. 208
  8. line 204-211 Table 9 and following lines: Mn, Mw/Mn, Tg, Tm, ΔHm, Tc, ΔHc should be corrected to Mn, Mw/Mn, Tg, Tm, ΔHm, Tc, ΔHc  - Corrected and Highlighted - Line No. 213-220
  9. line 299: abbreviations for ortho-, meta-, para- should be in italics, change m-Cresol to m-Cresol. The n-Methyl pyrrolidone is improbable (n- prefix means a linear chain, while N-Methyl pyrrolidone is a common compound (see Letter symbols for elements are italic when they are locants in chemical-compound names indicating attachments to heteroatoms, e.g. O-, N-, S-, and P- in the document referred in the first review round). However, also an opposite recommendation can be found in case of heteroatoms; capital and not lowercase N is anyway most probably what should be here. - Corrected and Highlighted - Table 11, Line No.308
  10. lines 299-303: I suggest to remove superscripts 1, 2, 3, 4; they look like solubility parameters powered to those values. It is misleading. The explanation without them is clear -  Corrected and Highlighted - Table 11, Line No.308-313
  11. line 299 Table 11: Right (decimal) alignement would be more reader-friendly. Corrected and Highlighted - Line No. 313
  12. line 320 Table 13: Adding of the temperature units is appreciated. However, the meaning of the other numbers has to be searched in the text around. The table legend or header should tell the reader what the table presents. The Effect is not a physical quantity. The values are amounts of the migrated components in water, probably divided by the sheet area. This (or, if I understood wrongly, their correct meaning) should be stated in the heading and the legend. In addition, the sheet thinkness is to be indicated here. The currently presented values are too uncertain numbers. In addition, a and b superscripts are misleading, they should be removed. - Corrected and Highlighted - Table 13 - Line No. 332
  13. line 326: The reference would be better immediately after the name of reported author. Similarly in lines 351, 353, 373, 379, 393, 464, 466, 557. - Corrected and Highlighted - Line No. 363, 365, 376, 401, 407, 411, 422, 423, 436, 488.
  14. line 388, 389: Insert space between number and units. -Corrected and Highlighted. Line No. 390 and 392.
  15. line 396: Change TiO2 to TiO2 here and through the whole text - Corrected and Highlighted. Line No. 424.
  16. line 432 Fig 4: If both reaction and aparature diagram are given in the figure, the reaction relating to each reactor should be graphically assigned. It seems to be space enough to increase the descriptions of apparature components to make them better readable. –A higher definition image was uploaded including text descriptions.
  17. line 438: The misleading comma persists, although its removal was reported. Corrected and Highlighted - Line No. 488.
  18. line 451: Space after fullstop still missing. Corrected and Highlighted - Line No. 501.
  19. line 472: Can you make the table wider and unwrap the rows? - Corrected and Highlighted - Table 16. Line No. 535.
  20. line 600: Where is the end of double-quoted text? - Corrected and Highlighted - Table 16. Line No. 648-649.
  21. Authors have added several references in the least laborius way, so that they have added a new paragraph just before conclusion and they need not renumber any references. Adding any missing information in the review article is welcome, however I feel that authors prefer to satisfy formally requierements of the reviewer rather than to rectify the demerits consisting in the too narrow selection of reported literature. This selection is appropriate for an original research article with the own results of authors. However, the review article without new results should, in my opinion, cover existing literature in a larger extent. The following references have been added:

https://doi.org/10.1002/pen.25623, Ref [109]
 https://doi.org/10.1016/j.eurpolymj.2007.06.045, Ref [117]
 https://doi.org/10.3390/polym11071193, Ref [106]
 https://doi.org/10.22141/2224-1507.9.1.2019.163056 Ref [110]
 https://doi.org/10.1016/S0141-3910(02)00372-5 Ref [51]
 https://doi.org/10.1007/s10439-015-1455-8 Ref [113]
 https://doi.org/10.1016/j.jmbbm.2019.103510 Ref [111]
 https://doi.org/10.1016/j.matdes.2017.11.031 Ref [114]
 https://doi.org/10.1016/j.addma.2020.101414 Ref [84]
 https://doi.org/10.1002/app.47824 Ref [43]
 https://doi.org/10.1002/masy.201900064 Ref [85]
 https://doi.org/10.3390/polym11091496 Ref [120]
 https://doi.org/10.1021/acs.iecr.7b00930 Ref [40]
 https://doi.org/10.3390/su12020652 Ref [95]
 https://doi.org/10.3390/app10020509 Ref [42]
 https://doi.org/10.3390/polym11121908 Ref [38]
 https://doi.org/10.14314/polimery.2020.7.8 Ref [112]
 https://doi.org/10.1016/j.compositesb.2018.04.017 Ref [96]
 https://doi.org/10.1002/pi.5588 Ref [97]
 https://doi.org/10.1007/s00068-020-01564-1 Ref [44]

  1. 99 is now in the reference list, but not used in the text. Please check what you have used from this paper and place it again.- Renumbered

Reviewer 2 Report

I accept the authors' responses. However, I believe that the properties of PLA determine its applications and in many cases, it is not a material capable of replacing the currently used materials. 

In the description of table 9  the  Mw/Mn – Polydispersity ratio 
should be replaced to Mw/Mn – dispersity index  

Author Response

Dear Reviewer,

Thank you very much for taking the time to review our article. We have made a range of changes to the article , seen in the new draft uploaded. Also to address your comments, we have made the following change: 

Table 9: Mw/Mn           –          Dispersity index

In the article we discuss the possibility of PLA to replace some existing polymers,    it is noted in the text that PLA possess very similar properties to PS, and may be a replacement option. Also in the text we describe routes to improve the functional properties of PLA with the use of additives, so that it could replace standard polymers for more applications. We hope this addresses the reviewers comments, and thank them again for their time and work on the article. 

Kind Regards,

The Authors

Reviewer 4 Report

After review the manuscript, this was improved with comments/observations done in previous version, only I recommend that highlight the relevance of work indicating in abstract that there is only few of applications of PLA-based lab ware due this is not indicated.

Author Response

Dear Reviewer,

Thank you for your comments, please see a new version of the article uploaded, with several changes, and reference updates. Based on your comments we have amended the abstract to add the following:    

"However, to-date only limited bioplastic replacement examples exist. In this review, common polymers used for labware are discussed, along with examining the possibility of replacing these materials with bioplastics, specifically polylactic acid (PLA). The material properties of PLA are described, along with possible functional improvements dure to additives. Finally, the standards and benchmarks needed for assessing bioplastics produced for labware components are reviewed."

We hope this address your comments, and we thank you for your time reviewing our article.

Kind Regards, 

The Authors